# Smart hybrid microscopy for cell-friendly detection of rare events

Willi L. Stepp [1] ✉, Giorgio Tortarolo [1], Juan C. Landoni [1],
Emine Berna Durmus [1], Santiago N. Rodriguez Alvarez[1], Kyle M. Douglass [1],
Martin Weigert [2,3] & Suliana Manley [1] ✉

Fluorescence microscopy offers unparalleled access to the spatial organization and dynamics of biological events in living samples, yet capturing rare processes over extended durations remains challenging due to trade-offs between exposure to excitation light and sample health. Here, we introduce hybrid-EDA, an event-driven acquisition (EDA) framework that combines the gentleness and contextual richness of phase-contrast with the functional specificity of fluorescence. We develop surveillance for events of interest in label-free microscopy using dynamics-informed neural networks that trigger smart fluorescence acquisitions upon detection. This allows us to dramatically reduce phototoxic damage while obtaining specific and functional information from fluorescence when beneficial. We demonstrate how hybrid-EDA enables improved imaging acquisitions of organelle contacts and mitochondrial divisions. We envision that hybrid-EDA will enable insights into a range of dynamic and rare biological processes, providing a powerful and general strategy for cell-friendly imaging.

Light microscopy is an invaluable tool for investigating the dynamics of biological systems, due to its non-invasive nature. Fluorescence-based contrast offers specificity by targeting dyes to molecules or structures of interest; on the other hand, label-free microscopy exploits the natural variations in the physical properties of cellular compartments and offers broader context[1]. Yet, these classical approaches have their limitations – fluorescence requires a high excitation irradiance and generates cytotoxic free radicals[2,3], while phase-contrast offers reduced phototoxicity but is insensitive to molecular details. Combining fluorescence and phase-contrast in correlative imaging provides both specificity and context, but maintains the phototoxicity of fluorescence microscopy.

Machine learning-based methods such as denoising and virtual staining have provided an alternative way of recovering fluorescence information with reduced phototoxicity. Denoising methods can learn a model of the noise to restore images[4–7], allowing data to be collected at a lower signal-to-noise ratio, corresponding to a lower light dose. Virtual staining uses paired fluorescence and label-free

images to train a model, which can then predict fluorescence patterns based on intrinsic contrast alone[8–10]. Thus, fluorescence-like images are generated at the low light dose of label-free imaging methods such as brightfield, differential interference contrast, or phase-contrast microscopy. However, this approach comes with major constraints: virtual staining can only reliably predict images when the label-free data contains signatures of features which should appear fluorescent. This is not generally the case for proteins or for functional readouts that are possible with fluorescent biosensors (e.g., $Ca^{2+}$, membrane potential).

Another approach, complementary to denoising, is smart microscopy. Previously, we developed event-driven acquisition (EDA)[11] to reduce light dose by driving our microscope with a closed-loop controller incorporating a deep learning-based detection module trained to predict specific, rare biological events. We demonstrated EDA with a detector developed to predict mitochondrial or bacterial division; this strategy allowed the microscope to toggle automatically from slow to fast imaging to capture in greater temporal detail the structural

[1]Institute of Physics, Swiss Federal Institute of Technology Lausanne (EPFL), Lausanne, Switzerland. [2]Center for Scalable Data Analytics and Artificial Intelligence (ScaDS.AI), Dresden, Germany. [3]TUD Dresden University of Technology, Dresden, Germany. ✉e-mail: willi@stepp.one; suliana.manley@epfl.ch

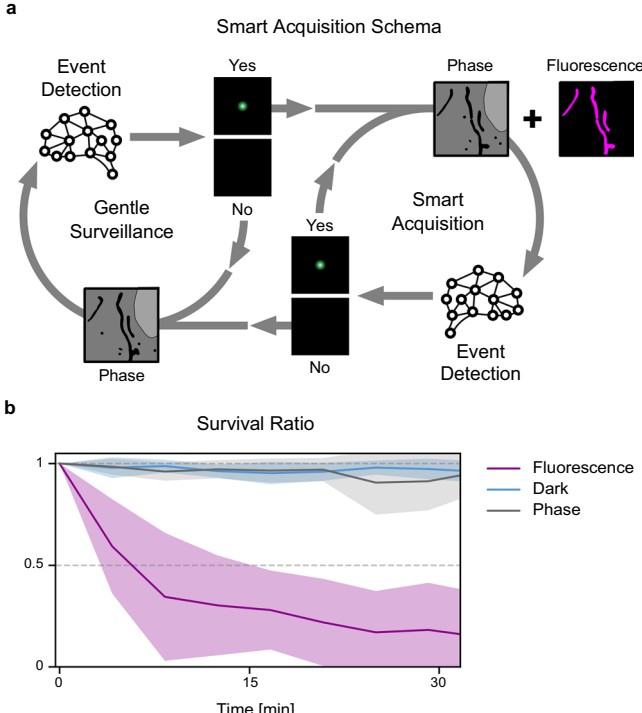

**a** Smart Acquisition Schema

Event Detection

Gentle Surveillance

Yes

No

Phase

Fluorescence

Smart Acquisition

Yes

No

Event Detection

Phase

**b** Survival Ratio

- Fluorescence
- Dark
- Phase

Time [min]

**Fig. 1 | Label-free detection of rare events balances trade-offs in live-cell hybrid-EDA. a** Schematic of the state-machine-like control of smart hybrid imaging. Neural network event detection is used to control the switch between gentle surveillance and smart correlative acquisition. **b** Survival ratio over time for different imaging modalities. Data were acquired at 1 fps and 100 ms exposure with typical illumination settings (Supplementary Note 1, Supplementary Fig. 1). Shaded lines represent mean ± SD. $n = 18$ measurements from 3 independent samples. Source data are provided as a Source Data file.

intermediates which accompany such divisions. Additional examples of real-time adaptive light microscopy measurements have been used to lower the light dose of widefield fluorescence[12], localization microscopy (MINFLUX)[13], stimulated emission depletion (STED)[14] or lattice light sheet microscopy[15]. However, the complementary combination of label-free and fluorescence modalities into a smart microscope remains unexplored, probably due to difficulties in distinguishing different subcellular features in label-free images, which makes event detection challenging.

Here, we report a hybrid method for detecting specific dynamic events in phase-contrast microscopy, and integrate it into a microscope to demonstrate real-time switching to fluorescence to capture the molecular and physiological signatures of mitochondrial contact sites and mitochondrial divisions. A key to reliable event detection in phase-contrast is including temporal information and memory in a specialized network architecture. Integrating this into the EDA paradigm allowed us to reduce the impact on sample health during surveillance by more than a 100-fold reduction in excess mortality, resulting in ten times longer experiments or, equivalently, capturing ten times more rare events. As an additional benefit, we could better exploit fluorescent sensors for live imaging despite their poor photostability.

## Results

A remaining limitation of EDA-like methods is their use of fluorescence to detect events; a significant proportion of the photon budget is spent during surveillance for event detection, leading to fluorophore bleaching, phototoxicity, and their impacts on cellular physiology. Our method transcends fluorescence-only EDA by combining the strengths of two complementary imaging techniques in hybrid-EDA: phase-contrast for gentle surveillance of the sample until an event of interest is detected, and consequent correlative phase and fluorescence imaging. Neural networks perform the challenging task of detecting events in content-rich phase-contrast images to trigger the modality switch on-the-fly. Effectively, we consider the microscope acquisition as a two-state machine (Fig. 1a), with one state corresponding to gentle surveillance in phase-contrast and the other state corresponding to smart multi-channel acquisition. During each state, the microscope acquires phase-contrast images, through which the biological system is continuously monitored by an event-detection neural network whose output determines when state transitions should be actuated. Details on the response latency between event detection and state transition are provided in the Supplementary Note 2.

Transmitted light microscopy is commonly accepted to be gentler than fluorescence microscopy[3], but we sought to quantify their differential impacts on cell survival during mitochondrial imaging. Mitochondria are exceptionally sensitive to phototoxicity[16], and thus serve as an important reference target for imaging. We acquired widefield fluorescence or phase-contrast images of mitochondria in COS-7 cells using typical acquisition parameters (Methods, Supplementary Note 1). To assess phototoxicity, we also measured nuclear envelope integrity, a hallmark of cell death, and observed that it was compromised in 50% of cells after only 5 min of fluorescence imaging at 1 frame-per-second (fps, Fig. 1b). By comparison, in phase-contrast acquisitions with comparable contrast, nuclear envelope integrity remains indistinguishable from 'no illumination' conditions, even after 1 h of imaging at 1 fps. Exponential decay fits of these curves reflect an ~50-fold decrease in half-life times and greater than 100-fold excess mortality of the cells in fluorescence imaging compared to phase-contrast (Supplementary Note 1, Supplementary Fig. 1). These data highlight that hybrid-EDA will be most beneficial for studying rare, dynamic events, where most time is spent in surveillance mode and fluorescence is only enabled rarely for short times.

### Organelle contact site detection by neural networks

Phase-contrast imaging naturally highlights cellular organelles due to differences in their molecular content relative to the cytoplasm, which gives rise to differences in refractive index. Organelles such as mitochondria, lysosomes, lipid droplets, and the endoplasmic reticulum can form heterotypic membrane contact sites, which play important functional roles in signaling, lipid metabolism, and organelle fission, among others[17]. However, many contacts are transient, lasting only seconds, and their frequency and function may be highly sensitive to cell health and state. Imaging at low phototoxicity is therefore paramount to obtain biologically relevant information. In phase-contrast movies, we observed dynamic contact events between different organelles of distinct tubular versus spherical morphologies. Based on fluorescence imaging, we identified the tubular organelles as mitochondria, and we hypothesized that the spherical organelles could be lysosomes or lipid droplets[18,19], both known to form functional contact sites with mitochondria. We therefore aimed to develop an event detection framework for such apparent contacts.

The main challenge in implementing hybrid-EDA is reliably predicting or detecting biological events, which is more challenging in label-free images than in fluorescence imaging. The visibility of diverse cellular structures in phase-contrast data effectively adds segmentation and classification tasks to event detection, whereas those tasks are circumvented by the specificity of the signal in fluorescence data.

To create a training dataset, we manually labeled contact events between mitochondria and spherical organelles in phase-contrast time-lapse images of COS-7 cells (Fig. 2a, Supplementary Note 3, Supplementary Fig. 2). For event detection, we trained neural networks with a U-Net architecture with the phase-contrast frames and a ground truth containing a local label consisting of a Gaussian located

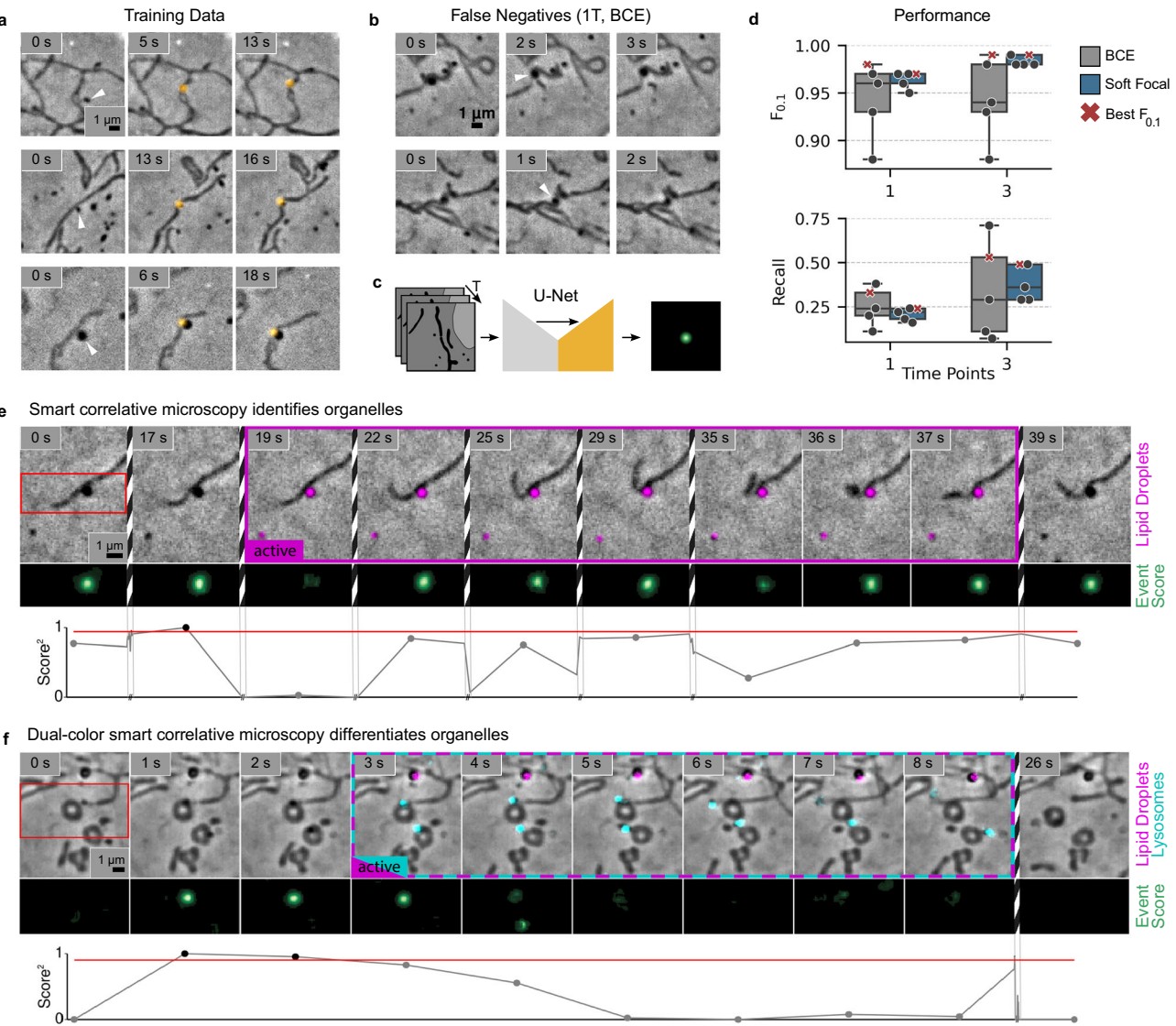

**Fig. 2 | Dynamics-informed U-Nets enable organelle contact detection.**
**a** Examples of positive events in the training data-set. **b** Examples of false negatives in the prediction of neural networks trained with a single-time-point input and binary cross-entropy loss. **c** Dynamics-informed U-Nets take several time points as input, and output a 2D event score indicating the locations of organelle contacts in the last time point. **d** $F_{0.1}$ and recall scores for neural networks trained with one or three time points and binary cross-entropy (gray) or soft focal loss (blue). One model with the best $F_{0.1}$ score is indicated with a red cross in both plots for each setting. **e** Smart hybrid acquisition of a lipid droplet-mitochondrion contact. The

event scores at the contact (green and bottom) exceed the set threshold, which triggers fluorescence imaging (magenta). Striped boundaries indicate frames that are not displayed here. The lipid droplet marker mEmerald-PLIN5 identifies the contacting organelle. **f** Dual-color smart hybrid acquisition of lysosomes (cyan) and lipid droplets (magenta). $N = 104$ total events and 5 individual training runs on the same training/evaluation data split for each loss and number of time points. Box plots mark the first quartile, median, and third quartile with the whiskers spanning the 5th and 95th percentile. Source data are provided as a Source Data file.

---

at each contact site. Alternative architectures such as detection-based models (YOLO[20], Spotiflow[21]) or transformer-based approaches (LW-DETR[22]) could be suitable for the event detection task. However, our choice of a U-Net as our backbone was motivated by several reasons: (1) computational efficiency for real-time feedback to the acquisition, (2) ability to provide precise spatial localization through skip connections, (3) efficiency with limited training data, and (4) ease of incorporating temporal information. The U-Net architecture has additionally been shown to be on par with more recent architectures in practical settings such as cell segmentation[23].

We adopted the $F_{0.1}$ score as our primary performance metric because it prioritizes precision (*P*, true positives over all predicted positives) over recall (*R*, ratio of actual positives recovered). In other words, it places an emphasis on reducing false positives. This is beneficial for phase-EDA, because the surveillance stage in phase-

contrast is exceptionally cell-friendly, while smart acquisition during a false positive exposes the sample to fluorescence imaging without benefit (1).

We first selected models with the best $F_{0.1}$ scores and used them to classify the events in the validation dataset. However, we observed a considerable spread in performance for models trained with identical parameters. To address this, we switched from cross-entropy to a soft focal loss function for training, a strategy known to be beneficial for the detection of rare positive events[24,25]. This stabilized the training at high performance scores. Yet, even these models yielded a high rate of false negatives—instances where genuine organelle contacts were missed (Fig. 2b). Therefore, we further improved our training approach by supplying a time series of images to the models in the form of additional image channels, enabling the networks to use dynamic information in the detection of events (Fig. 2c). With soft

focal loss, this increased the recall of the best performing models by almost a factor of 2, allowing more events to be captured during adaptive acquisitions (Fig. 2d, Supplementary Fig. 3). These improved models enabled efficient detection of apparent contact events between spherical organelles and mitochondria.

## Smart acquisitions of organelle contacts

We implemented a custom adaptive imaging framework based on pymmcore-plus[26] to control our microscope, which allowed us to incorporate our contact detection neural network to adapt acquisition parameters on-the-fly (Methods, Supplementary Note 4, Supplementary Fig. 4). We imaged COS-7 cells in phase-contrast with live surveillance. To mark lipid droplets, cells were transfected with perilipin 5 (PLIN5) fused to mEmerald fluorescent protein. Once a contact event was detected, the framework automatically switched the microscope to correlative phase and fluorescence imaging. This allowed us to identify the contacting spherical organelle as a PLIN5-positive lipid droplet (Fig. 2e, Supplementary Fig. 5a, b, Supplementary Movies 1–3).

While many detected contact partners were indeed lipid droplets, we also observed contacts with spherical organelles that were not PLIN5-positive. We therefore performed hybrid-EDA on cells stained with LysoTracker Red to mark lysosomes and LipidTOX to mark lipid droplets (Fig. 2f, Supplementary Fig. 5c, Supplementary Movies 4, 5). This enabled us to capture and distinguish mitochondria-lipid droplet and mitochondria-lysosome contact sites simultaneously, an otherwise unreliable task in phase-contrast, requiring three-color fluorescence imaging. By using the phase-contrast data to discern the mitochondrial morphology, we eliminate the need for an additional fluorescence channel and marker, enhancing the value per phototoxic effect for every channel. During labeling we identified one contact event on average every 3.1 min, with a duration of 20 s; thus, when imaging at 1 Hz hybrid-EDA captured 20 fluorescence frames per event instead of about 186 which would have been captured during continuous fluorescence. This represents a -nine-fold reduction in phototoxicity per event, or a nine times longer accessible experiment duration. Fluorescent dyes and proteins have developed to be increasingly efficient and photostable; improvements in functional reporters are less forthcoming, perhaps because their specificity in sensing constrains optimizing other attributes beneficial for imaging. TMRE (Tetramethylrhodamine ethyl ester) is widely used to measure mitochondrial membrane potential. However, the membrane potential itself is very sensitive to the cell's stress state[27]—meaning that even the moderate light exposure required for fluorescence imaging can induce changes. This sensitivity makes it challenging to reliably capture rapid fluctuations in membrane potential which are critical for mitochondrial quality-control, especially those which accompany mitochondrial dynamics. Inspired by recent reports showing differential bioenergetics in mitochondria interacting with lipid droplets[28,29], we sought to explore the rapid dynamics of membrane potential during transient interactions between mitochondria and lipid droplets. We detected short-lived (1–3 s) membrane potential fluctuations and flickering during our acquisitions, but did not find differential behavior between the mitochondria in contact with the lipid droplet and others in the vicinity (Supplementary Figs. 5d, 6, Supplementary Movie 6)—in contrast with reported data[28]. Combining the low phototoxicity of phase-contrast with the specificity of fluorescence markers and sensors in this way allows us to study organelle contacts under conditions closer to unperturbed cellular physiology, leading to more biologically relevant readouts, and with additional targets per acquisition.

## Detection of mitochondrial divisions in phase-contrast

Mitochondrial fission mediates organelle proliferation and quality control, and is thus a crucial process in maintaining cellular homeostasis[30]. Fission precursors in the form of transient constrictions are subtle, and may reverse without culminating in complete division

into two organelles[31]. To tackle the challenge of detecting pre-fission constrictions by phase-contrast microscopy, we needed to employ a more complex neural network training strategy (Fig. 3a). We first adapted the training data from[11] that contained ad hoc labels for candidate mitochondrial fission precursors based on two-channel images of mitochondria and the fission factor dynamin-related protein 1 (DRP1). We refined the labels using the dynamic information inherent in the time-lapse data. Specifically, we quantified frame-to-frame fluctuations to assess label accuracy and retained only those labels that consistently showed an event probability over 36%. With these filtered labels, we trained a U-Net to detect constrictions in single-channel images of the mitochondria. We then applied this network to the fluorescence channel of correlative time-lapses—combining mitochondrial fluorescence and phase-contrast—to identify potential division sites. Finally, we manually confirmed actual divisions and trained an additional U-Net with the verified labels and corresponding phase-contrast images (Fig. 3b).

Even with this multi-step training strategy, and when using multiple time points and soft focal loss as we had applied to detect organelle contacts, classifications of the validation data comprised significant false positives. Notably, long-lived false positives, mostly corresponding to cell membrane features such as folds or ruffles (Fig. 3c), posed a major problem by triggering hybrid-EDA over extended periods. This rendered the models unfit for informing our adaptive acquisitions.

Recognizing that the structural signatures of mitochondrial divisions are even more subtle than those of organelle contacts, we refined our approach to make more use of their dynamic signatures by developing a stateful U-Net. This architecture incorporates a state-capturing long short-term memory (LSTM) layer at the bottleneck between the encoder and decoder, equipping the model with temporal memory to track subtle, time-dependent features across image sequences (Fig. 3d)[32]. Using this architecture we achieved consistent improvement of model performance. As with organelle contacts, $F_{0.1}$ scores and precision on the validation data improved with more input time points (Fig. 3e, Supplementary Fig. 7). Recall values remain relatively low, but we observe no specific characteristics in the missed events (e.g., peripheral vs central divisions), implying that the sampled true positive divisions are unbiased in this regard (Supplementary Fig. 8, Supplementary Movies 7–10).

## Smart acquisitions of mitochondrial divisions and constrictions

DRP1 assembles at pre-constrictions, and actively further constricts mitochondria to drive membrane scission. Its low abundance and dynamicity at individual division sites makes DRP1 a challenging target for fluorescence imaging. Within seconds after division is complete, mitochondria typically recoil away from the fission site, while DRP1 disassembles, leaving as an open question whether it shares severing mechanisms with other dynamin family members[33–35]. We imaged COS-7 cells expressing DRP1-mEmerald in phase-contrast and detected pre-constrictions using the aforementioned networks. Switching on fluorescence allowed us to study the dynamics of DRP1 during and immediately after divisions (Fig. 3f, Supplementary Movie 11). We found that DRP1 signal is present at the tips of both daughter mitochondria for most of the divisions (Fig. 3g), distinct from dynamin-mediated fission of endocytic vesicles which occurs at the edge of the dynamin coat[34]. With an estimated division rate of 1 per 3.2 min for a typical cell, when imaging at 1 Hz we could image ten times longer in hybrid-EDA by imaging 20 fluorescence frames per event instead of about 192 if we were imaging continuously in fluorescence. We applied our smart imaging approach to measure membrane potential in the vicinity of constriction sites and at divisions, and captured events of temporal membrane potential loss—or 'flickering'—associated with these sites (Fig. 3h, Supplementary Fig. 9, Supplementary Movies 12–16). While

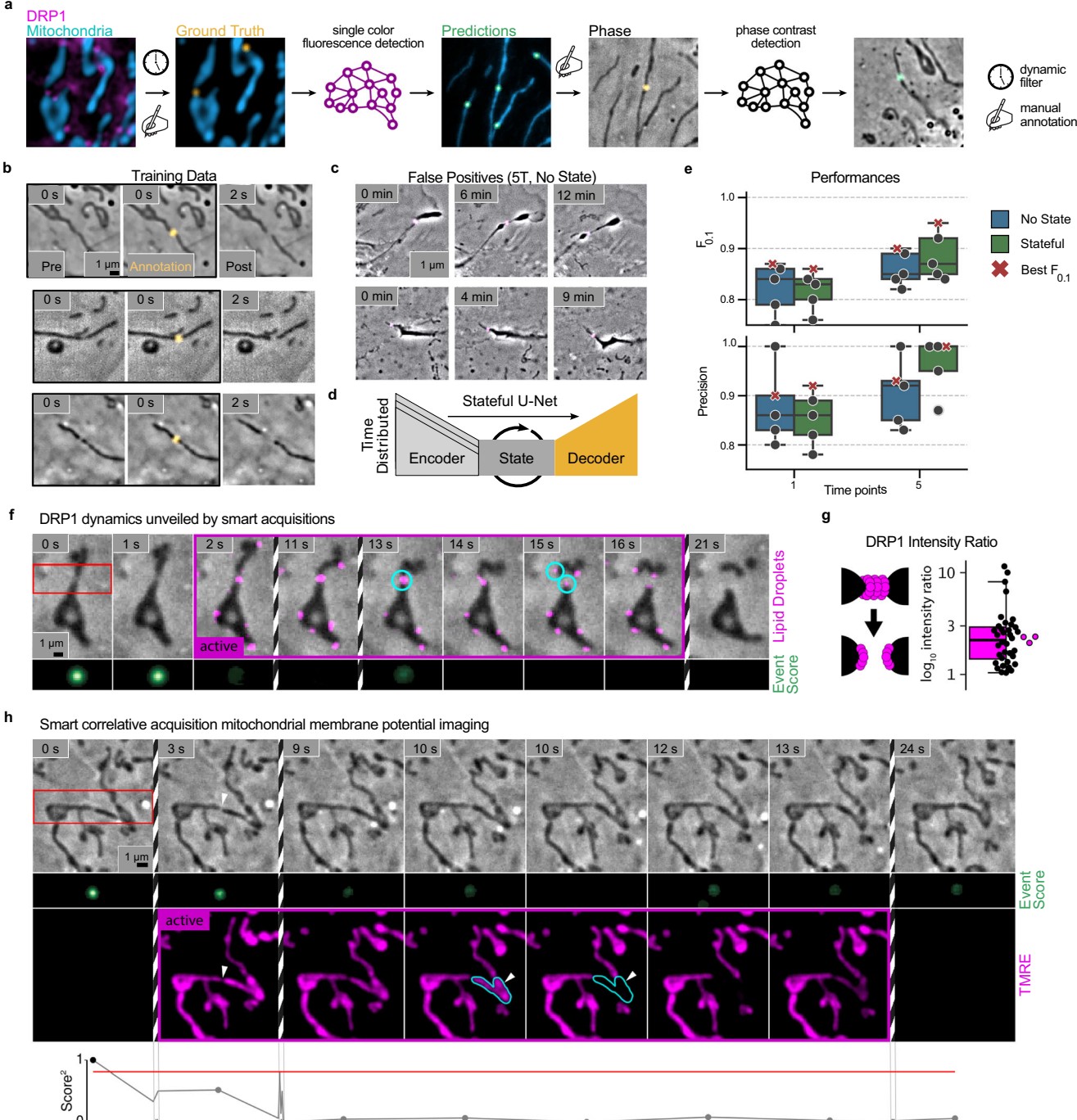

**Fig. 3 | Stateful U-Nets allow detection of mitochondrial division precursors in phase-contrast. a** Labeling approach for mitochondrial division pre-states in phase-contrast data. **b** Subset of frames of positive training events for divisions. **c** False positives in predictions from neural networks trained with five time points and soft focal loss, but no state layer in the bottleneck. **d** Architecture of a stateful U-Net with a time-distributed encoder to carry dynamic information to the bottleneck and a state layer with memory in the bottleneck. **e** Performance scores for neural networks with (green) and without (blue) state and trained on one and five time points. One model with the best $F_{0.1}$ score is indicated with a red cross in both plots for each setting. **f** Smart hybrid acquisition of a mitochondrial division with DRP1 fluorescence showing the dynamics of DRP1 disassembly and mitochondrial dynamics. Cyan circles mark division site before and the considered mitochondria daughter tips after division. **g** Schema of DRP1 constitution before and after division and ratios of DRP1 intensity on the tips of the two daughter mitochondria after division ($N = 36$ divisions across 3 independent experiments). Magenta points indicate the average intensity ratio per independent experiment. **h** Smart hybrid acquisition of membrane potential dynamics at a mitochondrial constriction that prevents potential propagation. White arrows mark the division site. Cyan polygons highlight the outline of the relevant part of the mitochondrion before and after membrane potential loss. $N = 665$ total events and 5 individual training runs on the same training/evaluation data split for each architecture and number of time points. Box plots mark the first quartile, median, and third quartile with the whiskers spanning the 5th and 95th percentile. Source data are provided as a Source Data file.

whole-mitochondrion flickering is readily observed with traditional microscopy[36], this rapidly reversible subdomain loss of potential would be undetectable at the lower imaging rates typically used in these experiments to avoid reporter photobleaching. Among a total of twenty division events, we captured 4 with significant fluctuations in membrane potential, only one of which was observed to reverse constriction without going to full separation of the organelles.

## Discussion

A central challenge in implementing hybrid-EDA lies in developing an event detection deep learning network of sufficiently high performance to enable the desired biological experiments. We find that detecting different biological events may require different training approaches and different neural network architectures. An essential common feature for phase-contrast detection is the inclusion of temporal information—we achieved this through multi-time-point input, which enabled the detection of organelle contacts, and the addition of a state-capturing layer that was necessary to detect more subtle fission precursor states in mitochondria. Including state into the architecture of neural networks has been beneficial for other image analysis problems such as video object segmentation with state space[37], or recurrent[38] models or cell cycle state prediction from brightfield images[39]. This approach may improve other bioimage analysis applications involving subtle, dynamic features, including commonly-used processing steps such as segmentation, denoising, or deconvolution of time-lapse data, depending on the context.

Augmenting deep learning-based detection models with a temporal state yields a more robust and accurate framework for event detection on the microscope. With this, hybrid-EDA can greatly benefit the measurement of rare cellular events and short-lived structures. For instance, even the well-studied processes of fission and fusion involved in mitochondrial network dynamics are intermittent and occur on the timescale of a few seconds. Further elusive events regulating organelle homeostasis include vesicle generation, mitochondrial DNA release, and pearling[40,41]. New approaches are needed to investigate the rapidly growing list of molecular partners involved in these processes[19,42,43] while reducing phototoxicity, which can impact their rates and mechanisms. In combination with biosensors and super-resolution microscopy, hybrid-EDA can help to enable the detailed investigation of the ultrastructure of rare mitochondrial complexes and facilitate the detailed investigation of their functional consequences[30]. Beyond mitochondria, hybrid-EDA can drive investigations of processes such as lipid droplet[44] and filopodium biogenesis[45] or cell cycle-linked events[39], especially in sensitive cells such as stem cells or primary neurons.

Smart microscopy, while greatly beneficial, comes with limitations and challenges. A large enough training dataset has to be gathered and annotated for event detection, an especially challenging task for rare events. The assessment of model performance for use in adaptive acquisitions is not straightforward; architecture optimizations can be necessary depending on the event specifics, and biases in the model may translate into biases in the resulting hybrid-EDA data. Finally, events of interest need to have a detectable and distinct signature in the surveillance technique, limiting the range of biological phenomena suitable for detection by phase-contrast alone. One way to mitigate this limitation would be to validate low-confidence predictions with brief exposure to fluorescence, after which the state switch would be executed.

The core idea behind hybrid-EDA can be further extended to increase the resulting benefits, for example by correlative imaging with higher resolution fluorescence imaging methods like structured illumination or STED. Tailoring the surveillance method to the events of interest could also expand the space of detectable events; for example, some cellular features may be detectable by an adaptive phase-contrast technique employing spatial light modulators[46]. Quantitative phase imaging adds another avenue for improvement of event detection by encoding physical characteristics and identity of components more reliably into intensity information. Extensive reconstruction procedures that are necessary for human observation and classification (e.g., in holotomography or ptychography) may not be necessary for interpretation by neural networks, because they might be able to extract information directly from the inputs. Collectively, these strategies pave the way for hybrid-EDA approaches that not only broaden the scope of detectable events but also streamline image processing, thereby offering a versatile platform for probing dynamic processes in a more cell-friendly way.

## Methods
### Sample preparation
African green monkey kidney cells (COS-7, RRID:CVCL_0224; ECACC: 87021302) were cultured in Dulbecco's modified Eagle medium (DMEM; ThermoFisher Scientific, 31966021) supplemented with 10% fetal calf serum (FCS) in glass-bottomed cell culture dishes (ibidi, 81158). COS-7 cells were obtained commercially and not authenticated in house. mEmerald-PLIN5 was a gift from S. Cohen. DRP1-mEmerald was cloned from DRP1-mCherry, a gift from G. Voeltz. For lipid droplet imaging, mEmerald-PLIN5[47] was transfected using Lipofectamine 2000 (ThermoFisher Scientific TFS, 11668030) and Opti-MEM (TFS, 31985062) following manufacturer's instructions (150 ng of plasmid per dish), and allowed to incubate for 12–24 h prior to imaging. For dual-color imaging of lipid droplets and lysosomes, cells were incubated with HCS LipidTOX™ Deep Red Neutral Lipid Stain (1:1000, Invitrogen, H34477) and LysoTracker™ Green DND-26 (50 nM, Invitrogen, 1020097) for 30 min before imaging. For DRP1 imaging, cells produced mEmerald-DRP1 or DRP1-mCherry after transfection as described above, and the latter were co-stained with a 1:10000 dilution of SYBR™ Gold Nucleic Acid Stain (Thermo Fisher Scientific, S11494) for 5 min, followed by two washes with transparent media. TMRE imaging was performed after 10 min incubation with 500 nM TMRE and three washes. In all cases, FluoroBrite DMEM (TFS, 1896701) was used for washes and during imaging. For toxicity measurements, cells were incubated with SYTOX-orange (100 nM, TFS, S11368), Hoechst 33342 (0.2 μg/ml, TFS, 62249) and MitoTracker Green FM (500 nM, TFS, M46750) for 10 min and washed three times before imaging.

### Microscopy
All imaging was performed on a Zeiss AxioObserver7 with a Plan Apochromat 63x/1.40 Oil Ph 3 objective for both phase-contrast and fluorescence imaging. A CoolLed pE-800 module was used for fluorescence illumination. Images were captured on a Photometrics Prime camera and cells were kept at 37 °C and 5% CO2. Adaptive acquisitions were performed using a custom software built on the pymmcore-plus environment described below. For toxicity measurements, we collected two data points for each imaging modality (fluorescence, phase, dark) in the same sample at different positions one after the other. This reduces effects from sample-to-sample variations in the survival data. Toxicity data was collected in three different samples, resulting in six data points for each laser power. A 553 high-pass filter was added to the brightfield illumination path for phase-contrast imaging. All four channels (SYTOX, Hoechst, MitoTracker, and phase-contrast) were recorded every 250 s. In between, the channel corresponding to the mentioned condition (MitoTracker, phase-contrast) was imaged at 1 fps with 100 ms exposure time. Phase-contrast imaging was performed at maximal power of the transmission illumination, while MitoTracker imaging was performed at 2% and 10% laser power (8 and 35 mW/mm², respectively).

### Neural network training
Organelle contact detection was trained on 111 events (58 positive, 53 negative), and evaluation data consisted of 29 events (18 positive, 11

negative). 596 events (319 positive, 277 negative) were used to train the detection of mitochondrial divisions with 157 events (86 positive, 71 negative) in the validation set. Training and evaluation data were split on an event basis upon registration in the event database, thus ensuring consistency of the training runs and performance scoring.

The last five frames of positive events were labeled with a Gaussian spot at the site of the event (division/contact). Organelle contact events were manually annotated based on a visible interaction of the two organelles by deformation of the mitochondrion. Additionally, both spatial proximity and temporal persistence (no visible separation for at least three frames) were considered for event selection. The divisions were followed over time and only the divisions with clear separation of the daughter mitochondria were labeled as fission events. The division site was tracked back in time and labeled as long as the future division site was distinguishable as a constriction or obvious by the mitochondrial morphology. All events used for training were checked by at least two separate people to ensure the reliability of the data-set. In a way to optimize training results, we used early models (trained on fewer events) on phase-contrast data, corrected false positives and false negatives, and added these events to the training data for subsequent training.

All networks use these five consecutive time points as input. labeled networks were trained with single images and the corresponding ground truth frames, while networks for multiple time points got the corresponding number of images plus a single ground truth frame corresponding to the last image in the time series.

Standard U-nets are of depth 2 with 16 initial filters, stateful variants used the same structure with a recurrent ConvLSTM2D layer at the bottleneck to capture spatiotemporal dependencies across the image sequence. Training was performed using TensorFlow/Keras 2.10 on NVIDIA Tesla T4 GPUs.

We used the $F_\beta$ score, which is a function of precision $P$ and recall $R$, with $\beta = 0.1$ as the main metric for performance analysis.

$$F_\beta = \frac{(1 + \beta^2) P \times R}{\beta^2 P + R} \tag{1}$$

This metric prioritizes the precision over the recall, and we chose it to reduce triggering on false positives. The threshold for classification was therefore optimized for the best $F_\beta$ scores and recall and precision are reported at that threshold unless declared otherwise. Prioritizing precision over recall results in the omission of some true events (false negatives), while reducing phototoxicity that would result from fluorescence imaging of false positives. This is consistent with the goal of Hybrid-EDA, to reduce phototoxicity, and the use of phase-contrast which is a nearly non-toxic monitoring modality. Missed events in validation datasets were visually inspected to and found not to introduce any evident systematic bias, and were used to fine-tune the models to increase recall.

### Microscope control
We built on the capabilities of the open-source microscope control software environment pymmcore-plus to allow for adaptive acquisitions. Hardware control, image capture, and storage are performed by components of the pymmcore-plus framework. Captured phase-contrast images are transferred to an analyzer component by the integrated event system. The analyzer stores a subset of time points, runs inference of the model on this data, and forwards the result to an interpreter that implements memory and thresholding. A threshold on the event score was implemented for switching to and from correlative imaging as well as a minimum number of correlative frames to be imaged after a trigger. If a modality switch is requested, a custom scheduler is notified to adapt the active acquisition settings. This scheduler supplies acquisition events for one or the other modality (phase-contrast/correlative) to the pymmcore-plus acquisition engine

at the correct times for the set frame rate (Supplementary Note 4). The execution of these events results in new phase-contrast data that is transferred to the analyzer to close the loop.

### Data analysis
SYTOX-orange entry into the nucleus was measured for detecting the onset of cell death. The Hoechst channel was used to segment nuclei and both SYTOX and Hoechst were normalized on a local region around two times the size of the nuclei. The ratio of the normalized intensities in the nucleus regions was used to determine the cell state. A ratio of 0.5 was used as a cutoff for the onset of cell death per cell.

The determination of image quality parameters (signal-to-noise ratio, intensity ratio, and contrast) was realized by marking several mitochondria per field of view and adjacent regions that didn't show any features of high contrast in brightfield and appeared dark in the fluorescence channel. The signal to noise ratio was defined as the signal in the mitochondria region divided by the standard deviation of the signal in the background region.

The analysis of mitochondrial membrane potential fluctuations during lipid droplet contacts was performed on captured event crops using CellProfiler 4.2.8[48]. Briefly, mitochondria were segmented from the TMRE channel using the IdentifyPrimaryObjects module (adaptive minimum cross-entropy, intensity-based separation), measuring their position and mean TMRE intensity, and tracking them (TrackObjects– Overlap method). Then, these mitochondria were classified between LD-contact positive and negative based on whether they were within 1.5 μm of the event detection center at the detection timepoint.

For the analysis of DRP1 abundance on mitochondria after division, regions of interest of $3 \times 3$ pixels were defined around the resulting tips. Integrated intensities were calculated after the subtraction of the camera offset value and translated into the corresponding ratios by dividing the higher intensity by the lower one.

### Statistics and reproducibility
No statistical methods were used to predetermine sample size. Sample size was chosen to achieve statistical significance, according to our previous experience and other studies from the field. Studies were performed on cells originating from the same cell line batch and randomly assigned to experimental conditions, and repeated on independent experimental days to ensure replicability. Data was excluded only upon signs of technical failure (poor imaging or staining quality) or cellular/mitochondrial stress (fragmentation, immobile mitochondria, apoptosis). When possible, analysis was performed in an automated manner to avoid bias.

### Reporting summary
Further information on research design is available in the Nature Portfolio Reporting Summary linked to this article.

## Data availability
Source data used in this study are provided in the Source Data file, and are available in the related Zenodo repository[49]. Source data are provided with this paper.

## Code availability
The training code used in this project is available at the LEB-EPFL GitHub page[50].

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

## Acknowledgements

We thank Hélène Perreten for technical support with cell culture and plasmid generation, and Sarah Cohen for supplying the PLIN5 construct. We are grateful to Talley Lambert for his work on the pymmcore-plus framework and helpful discussions for our implementation of the adaptive imaging framework. We acknowledge important input from Dafni Roumba, Adrien Marchandou, Elio Moreau, Emily Niemand, and Katarzyna Glinka. This work was supported by the Swiss National Science Foundation project grant (SNSF, 310030_215737 S.M.), the Human Frontiers in Science Program (HFSP, RGP0038/2021, S.M. and LT0041/2022-C, G.T.), the European Union's H2020 program under the European Research Council (ERC, CoG 819823 Piko, S.M.), and the Zeiss-EPFL Research IDEAS initiative (S.M.).

## Author contributions

W.S. and S.M. conceived and designed the project. W.S. and S.M. supervised the project. W.S., E.B.D., and J.C.L. collected training data and labeled the events. W.S. and S.R.A. implemented the neural network training, architecture, and evaluation with help from M.W. W.S. implemented the adaptive imaging framework. Lipid droplet experiments were performed by E.B.D. Mitochondrial division acquisitions and analyses were carried out and analyzed by W.S., G.T., and J.C.L., who also prepared the figures together with K.M.D. W.S. and S.M. wrote the manuscript with contributions from all authors.

## Competing interests

The authors declare no competing interests.
