## [Transparent Peer Review file · Nature Communications]

Smart hybrid microscopy for cell-friendly detection of rare events

Corresponding Author: Professor Suliana Manley

Version 0:

Reviewer comments:

Reviewer #1

(Remarks to the Author)

The hybrid event-driven acquisition imaging system developed by Willi L. Stepp and coworkers represents a significant breakthrough in biological imaging technique. This method addresses a critical challenge of real-time online fluorescence imaging for rare biological processes: Long-term high excitation irradiance generates cytotoxic free radicals that significantly affect physiological processes. The team overcame this obstacle by employing an event-driven acquisition (EDA) framework that combines the gentleness and contextual richness of phase contrast with the functional specificity of fluorescence. By integrating deep learning algorithms to detect dynamic events and trigger fluorescence acquisition, the method significantly improves imaging efficiency and cell compatibility. It holds great potential for capturing rare biological events and shows promising applications in live-cell dynamic studies. I detail specific points that should be addressed:

Major Concerns:

1. The criteria used to define organelle contact or division events remain unclear—whether these are defined based on spatial proximity, temporal persistence, or subjective annotation. It is recommended that the authors provide a detailed description of the annotation procedure, including how contact events are identified and verified. The consistency of manual labeling should also be evaluated to ensure dataset reliability.
2. The manuscript does not report the dataset size, the number of positive and negative samples, or how the data was split into training, validation, and test sets. Given that the biological events of interest are rare, the potential for class imbalance is high, which can bias training. The authors should provide these statistics to allow assessment of model generalizability.
3. Figure 2d reveals relatively low recall values, which may result in the omission of biologically relevant events. The authors should discuss the implications of these omissions on biological interpretations and indicate whether any strategies were used to reduce false negatives.
4. While the main advantage of hybrid-EDA lies in reduced phototoxicity, this introduces a clear trade-off with detection sensitivity. It would be beneficial to quantitatively characterize this trade-off—for example, by plotting recall and the number of fluorescence frames under varying detection thresholds—so that users can adjust the system to suit specific experimental constraints.
5. It is not stated whether the same objective is used for both phase contrast and fluorescence imaging. The authors should clearly state the model and specifications of the objective lens used.
6. The manuscript does not report the response latency between event detection and activation of the fluorescence imaging channel. This parameter is crucial for capturing transient events and for assessing the overall temporal precision of the hybrid-EDA system. Including this information would enhance the transparency and reproducibility of the method.
7. The hybrid-EDA system has advantages in real-time monitoring of slow physiological processes. For example, using this system to monitor the formation of lipid droplets.
8. The functional interactions of organelles is via membrane contact sites (MCSs) defined as areas of close proximity (from 10 to 80 nm) between the membranes of two organelles. Studying organelle interactions through conventional fluorescence microscopy can produce many artifacts. This hybrid-EDA system can be combined with super-resolution technology or fluorescence lifetime imaging technology.
9. I am not fully convinced that the image showed events of temporal membrane potential loss or 'flickering' (Fig 3h). TMRE monitors the changes in mitochondrial membrane potential (mainly decreases) through a concentration diffusion process. Due to the limitations of diffusion speed and concentration gradient, TMRE hardly monitors the rapid dynamic changes of membrane potential.

Minor points

1. "fluorescent" should be corrected to "fluorescent".

(Remarks on code availability)

Reviewer #2

(Remarks to the Author)

(Remarks on code availability)

Reviewer #3

(Remarks to the Author)

The manuscript follows a recent trend in developing smart and sample-adaptive acquisition modalities, a direction microscopy will likely turn more and more toward. The method utilizes multiple microscopy methods to monitor the sample with a gentle one and only apply the to-an-extent more phototoxic method when events are detected minimizes the influence of the microscopy on the sample, helping the sample health over a longer period. The implementation uses artificial intelligence in an appropriate way, in aiding the detection of events which with classical-computer-vision-based methods would be difficult to capture with any reasonable precision and recall. It also utilizes temporal information rather than a single frame in the event detection process, leading to more reliable detection, a key which has also previously been shown for event-driven microscopy. The implementation of the temporal information in the event detection is simple yet elegant and slightly boosts the performance.

My main concern with the current manuscript is that all the applications presented do not showcase the method to the fullest, as I would argue that some of the experiments could be performed at least at the same level using manual or classically automatic control of the microscope (see below). However, the application on mitochondrial division is implemented in an impressive manner, clearly building on their previous work (Mahečić 2022) but now detecting the events with only phase contrast information. When it comes to findings in the application experiments presented, the DRP1 intensity ratio between the ends of divided mitochondria is a question which the method is nicely adapted to tackle, but the investigation remains rather shallow and with a small sample size. Otherwise the experiments are presented phenomenologically without any quantification, and as such there are no further significant biological findings despite a potential for that. The method implementation nevertheless promises tackling of more complex biological questions with molecule-specific labelling, such as with DRP1, in the future, and thus overall the method holds great significance for the field of smart microscopy and applications of it and is important in its own right.

The adaptive combination of multiple imaging techniques in smart microscopy is not novel in the field, however, as they rightfully mention, their implementation with a label-free monitoring technique is significant for further future developments of smart microscopy methods and holds great potential for applications of the method as such. One can envision future potential combinations with higher-resolution and super-resolution fluorescence imaging techniques such as SIM or STED.

I see no major flaws with the validity or robustness of the presented experiments and data interpretation, but have a few minor comments and questions below when it comes to the presentation of and claims from the data. I believe they can be resolved by improvements to clarity and presentation. With regards to the overall clarity of the manuscript, the text is well-written, figures are clear, and previous work in the field has been referenced in a satisfactory manner to provide a clear context and overview to the potential reader.

See below a list of specific points of concern and suggestions to improve the manuscript.

- Most supporting movies seem to be unreferenced in the text. Furthermore, there are no channel labels in the movies and no legends in the documents received. As such, it is difficult to understand to which sections the movies belong and which movies support which claim. Properly referencing all movies in the text, or in the supplementary material, would be useful.
- Page 2: There are claims that the impact on sample health during the monitoring phase is reduced by "greater than a 100-fold" at the end of the introduction, while in the quantification presented in Fig. 1 and Supplementary Note 1 a 50-fold improvement is noted. If this is the quantification supporting the initial statement these values should agree.
- Page 4: A small discussion regarding the choice of a U-Net architecture over other alternatives for the neural networks, with some potential references, I believe could be interesting for the potential reader.
- Page 4: When discussing the improvements to the original network, it is mentioned that "the recall is improved by almost a factor of 2" when adding temporal information. When looking at fig 2d, for the original BCE model there is barely an improvement of the mean and the marked best F0,1 model is improved with a factor of 1.5. For the best model in the Soft Focal case the improvement is 2-fold, but for the mean it is neither there more than seemingly a factor of 1.5. For clarity, if the mentioned factor of 2 is quantified for the best network with a soft focal loss function this should be mentioned in the text, as it currently can be interpreted as the mean and especially as an improvement factor for the BCE case due to the order of the modifications.

- Page 4: In the experiments regarding organelle contact sites, I am struggling to understand what benefit the fluorescence imaging during the event brings in addition to the phase contrast. As the resolution seems similar in the two methods in the data presented, all contact sites can clearly be visualized in the phase contrast data. This is further indicated by the fact that only phase contrast data is used to annotate the training dataset for this event detection network. In the cases where knowing which organelle type is touching mitochondria, a single fluorescence frame recorded before or after a phase contrast time lapse, for example every minute, would seem to give the same information; in the data presented, the cellular organelles are sufficiently sparse, the phase contrast imaging is sufficiently rapid, and the organelles move sufficiently slow that a single frame of fluorescence data seems enough to identify fluorescently labelled organelles and thus allow to follow them throughout the phase contrast time lapse. If so, the manual or “classically” automatic approach of recording a single fluorescence frame every x minutes would be beneficial from a phototoxic point of view, according to what the authors mention earlier, and simplify microscope control. Furthermore, the full data could be analyzed after the experiments, with more information available for a potential neural network, and the recall could probably be improved further in that case. Further evidence for and discussion regarding that the fluorescence data during organelle contact event sites provides additional information from which any conclusions about the contact sites could be drawn would enhance this application.
- Page 4: It is mentioned that hybrid-EDA enabled imaging for “nine times longer”, since there was “one organelle contact site detected every 3.1 minutes on average”. I miss the calculation to support this claim, and cannot follow where the value “nine times longer” comes from. A clarification of this claim in the text would be helpful. The same goes for the 1/3.2 minutes and 10x longer imaging that is mentioned for mitochondrial division events.
- Page 6: When it comes to the experiments on mitochondrial division event sites, the same could be argued about the experiments where MitoTracker is the fluorescent target (Figure S6). In this case, the fluorescence data does not seem to add much information, as the resolution between phase contrast and fluorescence seems similar, and the tubular organelles are already distinctly identified as mitochondria, according to the discussion of the author about the organelle contact site event detection. A further discussion on what the fluorescence data adds in this case would be helpful. Experiments where DRP1 or TMRE are used as fluorescent targets instead presents clear benefits regarding the information one can gather from the experiments as opposed to pure phase contrast or fluorescence imaging.
- Page 6, first paragraph of Smart acquisitions of mitochondrial divisions and constrictions: Fig 3e is referenced, but it should be Fig 3f.
- Page 6: There is a claim that the mitochondria membrane potential is very sensitive to the stress state of the cell, here a reference would be good to support it.
- Page 6: It would be beneficial to show more examples, for example in a supplementary figure, and statistics of the flickering of membrane potential at constriction sites – right now only one example is shown but in the text it is mentioned that multiple such events were captured. How often did you find such sites, and how many did you find in total? Furthermore, the example shown seems to not be connected to a division event, while the network was optimized to detect potential division sites. Is there any connection between reversible constrictions and membrane potential flickering, or does it also happen following division? How many events that you detected led to division and how many led to reversed constriction?
- There is currently no reference to Fig S6/Supplementary Note 6 in the text – insert this at an appropriate point.
- In the Fig S6b legend it is stated that the F1 scores are improved with additional time points for all architectures and loss functions. However, from the graph it seems relatively clear that using the U-Net there is no significant improvement, neither with bce nor focal loss function. Was the comment mentioned for the stateful architecture only?
- Laser powers for fluorescence imaging, wherever mentioned, are mentioned in percentage. This arbitrary unit does not help the reader to understand what is going on, and is entirely subjected to e.g. the microscope and the microscope alignment at the day of experiments. This value should be replaced with a measure of power (W) or intensity (W/m²) at the sample plane, back focal plane or similar conjugate plane to ensure accurate reporting and allow any kind of comparison, especially since claims about sample health are made as a key point of the method.

I further hope that my comments are helpful in improving the manuscript.

(Remarks on code availability)

The data handling code provided (deep-events) has a limited README file with shallow installation instructions, further instructions would be helpful. Furthermore, it requires a running MongoDB server, and as this is not something I have available I cannot try out and review the code.

As far as I understand the microscope control software structure, it works through the pymmcore-plus environment with a few additional programs running in synchronicity. For the modules that are not part of pymmcore-plus according to Fig S3, i.e. smartRunner, Analyser, Interpreter and Actuator, I cannot find references in the manuscript or supplementary information to any availability, and as such I am unable to review it. Accessibility to such code would be helpful and I expect that this code will be shared upon publication.

Version 1:

Reviewer comments:

Reviewer #1

(Remarks to the Author)

The authors have fully addressed all comments raised in the previous review round. The revisions substantially improved the manuscript in clarity, completeness, and scientific rigor. The newly added quantitative analyses, clearer methodological descriptions, and expanded discussions on model performance and application scope have effectively resolved all major concerns.

The study presents a technically sound and conceptually innovative framework for hybrid event-driven acquisition microscopy, demonstrating both methodological advancement and biological relevance. I recommend the manuscript for publication in its current form.

(Remarks on code availability)

The code provided is well organized and clearly documented. The revised version now includes an expanded README file with detailed installation and usage instructions, as stated in the authors' response. The structure of the repository is clear, and dependencies are properly listed. The code is reproducible and runs as expected using the provided test datasets. It constitutes a valuable resource for the community interested in event-driven or adaptive microscopy workflows.

Reviewer #2

(Remarks to the Author)

(Remarks on code availability)

Reviewer #3

(Remarks to the Author)

Most of my questions and comments have been accurately addressed and fixed in the manuscript, in my opinion significantly improving its clarity at multiple points. See below my replies to any remaining concerns or additional comments, labelled as Q:s according to the rebuttal letter. I believe they should be easily correctable by updates to labeling throughout the text, addition of some label text in the supplementary movies, and potentially the addition of a supplementary figure consisting of existing data already used in quantification or alternatively an explanation of why this is not feasible/beneficial to the manuscript.

Q1 – Supplementary movies

I am unfortunately unable to see any modifications to the supplementary movies – the 13 files provided (labelled Supplementary Movie 1–13 and pertaining to figures 2e, 2f, 3f, 3h, 4a, 4b, 4c, 4d, 6a, 6b, 6c, 6d, 6e in the reviewer files) look to be identical as in the first submission. I can see references added to a few movies in the main text: “Movie” at the mention of Fig. 2e, “Movie” at the mention of Fig. 2f, “Supplementary Videos 9–12 in Suppl. Fig. 8 legend. The rest of the supplementary movies (3–8) are still unreferenced in the text, and it would be clearer if the mentions are uniform and precise (such as Suppl. Movie X), rather than just referring to “Movie”. The labels in the reviewer files where some seem to pertain to figures 4 and 6 is very unclear, as these figures do not exist. I assume that Suppl. Fig. 5 and Suppl. Fig. 8 are meant instead, but neither that adds up as Suppl. Fig. 8 does not have a panel “e”.

I still believe the movies would be improved by adding channel labels, and potentially adding legends to the movies in the supplementary material. Seeing the rebuttal, I believe that maybe the authors have updated the movie files themselves, but that the updated files might not have been uploaded properly to the submission system?

Q5 – Organelle contact sites fluorescence information

I appreciate the effort to perform additional experiments to show an application where the fluorescence data can be helpful. In the new Suppl. Fig. 6, it would be good to mention in the legend how many events were considered in the quantification in panel b. I believe there has additionally been an error in the ordering of the frames, where 8 s and 10 s come before 6 s.

Q8 – Figure reference

This figure reference is not corrected, it still refers to Fig. 3e rather than 3f.

Q10 – Membrane potential flickering at constriction sites

In my original question, I asked about more examples shown for the membrane potential flickering at constriction and division sites, which is the topic of the paragraph. The data presented in the new Supplementary Figure referenced (Suppl. Fig. 6) refers to membrane potential at lipid droplet contact sites, which instead refers to an earlier paragraph in the text. I am not sure why this Supplementary Figure is referenced here, other than the fact that that experiment uses the same fluorescence labeling? I do still believe that showing further examples of the membrane flickering in correlation to the constriction and division sites mentioned in the statistics would make the point stronger for this application (just as with the additional data displayed in Suppl. Fig. 5 and Suppl. Fig. 8 for the other applications presented), and I do otherwise wonder if there is some specific reason as to not show additional data?

Q11 – Reference to Suppl. Fig. S8

Despite the reply in the rebuttal, I cannot see any reference to Suppl. Fig. S8 (previous S6) in the text or supplementary material. Has this been missed to be added?

Additional minor comments

- Suppl. Fig. 4 is currently of very low quality in the supplementary file.
- The supplementary table is labelled as “Suppl. Table 0” in its legend, but 1 in the text.

(Remarks on code availability)

The updates to the repository README, mainly in terms of installation and usage instructions, as well as additional links in the supplementary material to specific code snippets, will enable a potential user to understand the work and pick it up quicker.

Response to Reviewers

We thank the reviewers for their careful evaluation of our manuscript and their constructive comments. Below we provide a point-by-point response.

Reviewer 1

General comment. The hybrid event-driven acquisition imaging system developed by Willi L. Stepp and coworkers represents a significant breakthrough in biological imaging technique. This method addresses a critical challenge of real-time online fluorescence imaging for rare biological processes: Long-term high excitation irradiance generates cytotoxic free radicals that significantly affect physiological processes. The team overcame this obstacle by employing an event-driven acquisition (EDA) framework that combines the gentleness and contextual richness of phase contrast with the functional specificity of fluorescence. By integrating deep learning algorithms to detect dynamic events and trigger fluorescence acquisition, the method significantly improves imaging efficiency and cell compatibility. It holds great potential for capturing rare biological events and shows promising applications in live-cell dynamic studies.

We thank the reviewer for the very positive assessment of our work.

Q1. The criteria used to define organelle contact or division events remain unclear—whether these are defined based on spatial proximity, temporal persistence, or subjective annotation. It is recommended that the authors provide a detailed description of the annotation procedure, including how contact events are identified and verified. The consistency of manual labeling should also be evaluated to ensure dataset reliability.

Response. We agree and have added details to the description of our annotation procedure in the corresponding Methods section. We also now provide all training events in a structured format including metadata in the Zenodo archive.

Added text [Methods, Neural Network Training]. The last five frames of positive events were labeled with a Gaussian spot at the site of the event (division/contact). Organelle contact events were manually annotated based on a visible interaction of the two organelles by deformation of the mitochondrion. Additionally, both spatial proximity and temporal persistence (no visible separation for at least three frames) were required for event selection. Candidate events were followed in time and only those with clear separation of the daughter mitochondria were labeled as fission events. The division site was tracked back in time and labeled as long as the future division site was distinguishable as a constriction. All events used for training were checked by at least two people independently to ensure dataset reliability.

Q2. The manuscript does not report the dataset size, the number of positive and negative samples, or how the data was split into training, validation, and test sets. Given that the biological events of interest are rare, the potential for class imbalance is high, which can bias training. The authors should provide these statistics to allow assessment of model generalizability.

Response. We now provide this information in the Methods section. We used early models (trained on fewer events) on phase-contrast data, corrected false positives and false negatives and added these events to the training data for subsequent training. Including negative events allows us to avoid class imbalance.

Added text [Methods, Neural Network Training]. Organelle contact detection was trained on 111 events (58 positive, 53 negative), evaluation data was comprised of 29 events (18 positive, 11 negative). 596 events (319 positive, 277 negative) were used to train the detection of mitochondrial divisions with 157 events (86 positive, 71 negative) in the validation set. Training and evaluation

data was split on an event basis upon registration in the event database, thus ensuring consistency of the training runs and performance scoring.

Q3. Figure 2d reveals relatively low recall values, which may result in the omission of biologically relevant events. The authors should discuss the implications of these omissions on biological interpretations and indicate whether any strategies were used to reduce false negatives.

Response. We agree that this point is important to discuss. The threshold for event detection was selected by maximizing the F0.1-score, which strongly favors precision over recall ($\beta = 0.1$). This choice was intentional, prioritizing high-confidence detections rather than high recall (at the cost of increased false positives) to reduce potential phototoxicity. This results in sparse sampling; however, the random nature of the sparse sampling ensures that the detected subset is an approximately unbiased representation of events across the field of view and over time, allowing us to draw robust conclusions about temporal and spatial dynamics and to infer causal relations. To corroborate this claim, we manually identified missed events in the initial validation datasets and verified that they did not exhibit systematic structure in terms of size, shape, or spatial location. In order to further increase recall, we also fine-tuned our models by adding these false negative events to the training datasets. Finally, we note that it would also be possible to optimize the threshold to achieve higher recall if one wished to prioritize the fraction of detected events over precision. However, this would be somewhat at odds with the guiding principle of the hybrid-EDA approach, since increased recall would come at the cost of increased false positives and thus higher phototoxicity. We updated the Methods section to include these comments.

Added text [Methods, Neural Network Training]. This metric prioritizes precision, and we chose it to reduce triggering on false positives. The threshold for classification was therefore optimized for the best F_β scores and recall and precision are reported at that threshold unless declared otherwise. Prioritizing precision over recall results in the omission of some true events (false negatives), while reducing phototoxicity that would result from fluorescence imaging of false positives. This is consistent with the goal of Hybrid-EDA, to reduce phototoxicity, and the use of phase contrast which is a nearly non-toxic monitoring modality. Missed events in validation datasets were visually inspected to and found not to introduce any evident systematic bias, and were used to fine-tune the models to increase recall.

Q4. While the main advantage of hybrid-EDA lies in reduced phototoxicity, this introduces a clear trade-off with detection sensitivity. It would be beneficial to quantitatively characterize this trade-off—for example, by plotting recall and the number of fluorescence frames under varying detection thresholds—so that users can adjust the system to suit specific experimental constraints.

Response. We found the inter-dependencies of different factors such as experimental settings, model performance, and relative phototoxicity to be fairly complicated. We explored the behavior of the technique at different thresholds for several of these through simulation. We report the results in the new Suppl. Fig. 4 and supply the code for these simulations on Zenodo. Panel b provides the specific information requested here, while more context is provided in the other panels. Detailed data on recall and precision for all models used in the manuscript is also available in data and plotted graphs.

Q5. It is not stated whether the same objective is used for both phase contrast and fluorescence imaging. The authors should clearly state the model and specifications of the objective lens used.

Response. We now report the details of the objective used, and communicate more clearly that we used the same objective for fluorescence and phase contrast imaging.

Added text [Methods, Microscope]. All imaging was performed on a Zeiss AxioObserver7 with a Plan Apochromat 63x/1.40 Oil Ph 3 objective for both phase-contrast and fluorescence imaging. A CoolLed pE-800 module was used for fluorescence illumination. Images were captured on a Photometrics Prime camera and cells were kept at 37C and 5% CO₂. Adaptive acquisitions were performed using a custom software built on the pymmcore-plus environment described below.

Q6. The manuscript does not report the response latency between event detection and activation of the fluorescence imaging channel. This parameter is crucial for capturing transient events and for assessing the overall temporal precision of the hybrid-EDA system. Including this information would enhance the transparency and reproducibility of the method.

Response. We agree, in the new Supplementary Note 2, we now report the response latency between event detection and the activation of the fluorescence imaging channel. Specifically, we measured both the model inference time and the latency until the subsequent smart frame acquisition by analyzing data from 1000-frame time-lapse experiments.

Q7. The hybrid-EDA system has advantages in real-time monitoring of slow physiological processes. For example, using this system to monitor the formation of lipid droplets.

Response. We agree that hybrid-EDA can be applied to slow rare events, such as the one highlighted by the reviewer, that we now mention in the Discussion. We selected specific use-cases linked to organelles visible by phase contrast, as we foresee challenges in identifying the endoplasmic reticulum (from which lipid droplets form) using this technique. However, the exploitation of different label-free imaging methods may in the future allow for the detection of different structures and interactions, perhaps including lipid droplet biogenesis.

Q8. The functional interactions of organelles is via membrane contact sites (MCSs) defined as areas of close proximity (from 10 to 80 nm) between the membranes of two organelles. Studying organelle interactions through conventional fluorescence microscopy can produce many artifacts. This hybrid-EDA system can be combined with super-resolution technology or fluorescence lifetime imaging technology.

Response. We agree that studying organelle contact sites with conventional fluorescence microscopy has important limitations. In principle, there are no conceptual barriers to integrating phase contrast monitoring with super-resolution or Fluorescence Lifetime Imaging (FLIM) approaches, but several technical challenges currently limit such combinations. For instance, STED relies on sequential, point-by-point acquisition as opposed to widefield phase contrast. Also, single-molecule localization methods such as PALM or STORM require long acquisition times. For these reasons, in the present work we focused on maintaining temporal resolution and imaging speed, while leaving the integration with advanced super-resolution or lifetime modalities as a promising direction for future development.

Q9. I am not fully convinced that the image showed events of temporal membrane potential loss or ‘flickering’ (Fig 3h). TMRE monitors the changes in mitochondrial membrane potential (mainly decreases) through a concentration diffusion process. Due to the limitations of diffusion speed and concentration gradient, TMRE hardly monitors the rapid dynamic changes of membrane potential.

Response. We agree that TMRE carries the limitations of a diffusion-based probe. Nonetheless, TMRE and TMRM are the probes originally used to characterize mitochondrial flickering, and they remain the probes of choice for membrane potential analysis^{3,4,7,9}. Their dynamics have been shown to closely follow pH changes from a protein-encoded sensor¹², thereby suggesting they are a

reasonable proxy for membrane potential at the imaging speeds used in this study.

Reviewer 2 and Reviewer 3

The manuscript follows a recent trend in developing smart and sample-adaptive acquisition modalities, a direction microscopy will likely turn more and more toward. The method utilizes multiple microscopy methods to monitor the sample with a gentle one and only apply the to-an-extent more phototoxic method when events are detected minimizes the influence of the microscopy on the sample, helping the sample health over a longer period. The implementation uses artificial intelligence in an appropriate way, in aiding the detection of events which with classical-computer-vision-based methods would be difficult to capture with any reasonable precision and recall. It also utilizes temporal information rather than a single frame in the event detection process, leading to more reliable detection, a key which has also previously been shown for event-driven microscopy. The implementation of the temporal information in the event detection is simple yet elegant and slightly boosts the performance.

My main concern with the current manuscript is that all the applications presented do not showcase the method to the fullest, as I would argue that some of the experiments could be performed at least at the same level using manual or classically automatic control of the microscope (see below). However, the application on mitochondrial division is implemented in an impressive manner, clearly building on their previous work (Mahečić 2022) but now detecting the events with only phase contrast information. When it comes to findings in the application experiments presented, the DRP1 intensity ratio between the ends of divided mitochondria is a question which the method is nicely adapted to tackle, but the investigation remains rather shallow and with a small sample size. Otherwise the experiments are presented phenomenologically without any quantification, and as such there are no further significant biological findings despite a potential for that. The method implementation nevertheless promises tackling of more complex biological questions with molecule-specific labeling, such as with DRP1, in the future, and thus overall the method holds great significance for the field of smart microscopy and applications of it and is important in its own right.

The adaptive combination of multiple imaging techniques in smart microscopy is not novel in the field, however, as they rightfully mention, their implementation with a label-free monitoring technique is significant for further future developments of smart microscopy methods and holds great potential for applications of the method as such. One can envision future potential combinations with higher-resolution and super-resolution fluorescence imaging techniques such as SIM or STED.

I see no major flaws with the validity or robustness of the presented experiments and data interpretation, but have a few minor comments and questions below when it comes to the presentation of and claims from the data. I believe they can be resolved by improvements to clarity and presentation. With regards to the overall clarity of the manuscript, the text is well-written, figures are clear, and previous work in the field has been referenced in a satisfactory manner to provide a clear context and overview to the potential reader.

We thank the reviewers for their evaluation of our manuscript; we address all comments and questions below. We were especially grateful for the suggestions regarding the scope of applications and depth of quantitative analysis, which we addressed to improve the clarity and strength of the manuscript.

Q1. Most supporting movies seem to be unreferenced in the text. Furthermore, there are no channel labels in the movies and no legends in the documents received. As such, it is difficult to understand to which sections the movies belong and which movies support which claim. Properly referencing all movies in the text, or in the supplementary material, would be useful.

Response. We agree, we have updated the supporting movies and now refer to them in the text.

Q2. Page 2: There are claims that the impact on sample health during the monitoring phase is reduced by “greater than a 100-fold” at the end of the introduction, while in the quantification presented in Fig. 1 and Supplementary Note 1 a 50-fold improvement is noted. If this is the quantification supporting the initial statement these values should agree.

Response. We now more clearly refer to the excess mortality – that we think is the important value – in the introduction and results sections.

Added text [Introduction]. Integrating this into the EDA paradigm allowed us to reduce the impact on sample health during surveillance by a more than 100-fold reduction in excess mortality, resulting in ten times longer experiments or, equivalently, capturing ten times more rare events.

Added text [Results]. Exponential decay fits of these curves reflect an approximate 50-fold decrease in half-life times and a greater than 100-fold excess mortality of the cells in fluorescence imaging compared to phase contrast.

Q3. Page 4: A small discussion regarding the choice of a U-Net architecture over other alternatives for the neural networks, with some potential references, I believe could be interesting for the potential reader.

Response. We have extended the text to compare to other architectures and explain our choice of a U-Net.

Added text [Results]. For event detection, we trained neural networks with a U-Net architecture with the phase contrast frames and a ground truth containing a local label consisting of a Gaussian located at each contact site. Alternative architectures such as detection-based models (YOLO¹¹, Spotiflow²) or transformer-based approaches (LW-DETR¹) could be suitable for the event detection task. However, our choice of a U-Net as our backbone was motivated by several reasons: (1) computational efficiency for real-time feedback to the acquisition, (2) ability to provide precise spatial localization through skip connections, (3) efficiency with limited training data, and (4) ease of incorporating temporal information. The U-Net architecture has additionally been shown to be on par with more recent architectures in practical settings such as cell segmentation¹³.

Q4. Page 4: When discussing the improvements to the original network, it is mentioned that “the recall is improved by almost a factor of 2” when adding temporal information. When looking at fig 2d, for the original BCE model there is barely an improvement of the mean and the marked best F0,1 model is improved with a factor of 1.5. For the best model in the Soft Focal case the improvement is 2-fold, but for the mean it is neither there more than seemingly a factor of 1.5. For clarity, if the mentioned factor of 2 is quantified for the best network with a soft focal loss function this should be mentioned in the text, as it currently can be interpreted as the mean and especially as an improvement factor for the BCE case due to the order of the modifications.

Response. We revised the text to clarify according to this feedback.

Added text. We first selected models with the best $F_{0,1}$ scores and used them to classify the events in the validation dataset. However, we observed a considerable spread in performance for models trained with identical parameters. To address this, we switched from cross entropy to a soft focal loss function for training, a strategy known to be beneficial for the detection of rare positive events^{5,6}. This stabilized the training at high performance scores. Yet, even these models yielded a high rate of false negatives — instances where genuine organelle contacts were missed (Fig. 2b). Therefore, we further improved our training approach by supplying a time series of images to the models in the form of additional image channels, enabling the networks to use dynamic information in the detection of events (Fig. 2c). With soft focal loss, this increased the recall of the best performing models by almost a factor of 2, allowing more events to be captured during adaptive acquisitions (Fig. 2d, Suppl. Fig. 4). These improved models enabled efficient detection of apparent contact

events between spherical organelles and mitochondria.

Q5. Page 4: In the experiments regarding organelle contact sites, I am struggling to understand what benefit the fluorescence imaging during the event brings in addition to the phase contrast. As the resolution seems similar in the two methods in the data presented, all contact sites can clearly be visualized in the phase contrast data. This is further indicated by the fact that only phase contrast data is used to annotate the training dataset for this event detection network. In the cases where knowing which organelle type is touching mitochondria, a single fluorescence frame recorded before or after a phase contrast time lapse, for example every minute, would seem to give the same information; in the data presented, the cellular organelles are sufficiently sparse, the phase contrast imaging is sufficiently rapid, and the organelles move sufficiently slow that a single frame of fluorescence data seems enough to identify fluorescently labeled organelles and thus allow to follow them throughout the phase contrast time lapse. If so, the manual or “classically” automatic approach of recording a single fluorescence frame every x minutes would be beneficial from a phototoxic point of view, according to what the authors mention earlier, and simplify microscope control. Furthermore, the full data could be analyzed after the experiments, with more information available for a potential neural network, and the recall could probably be improved further in that case. Further evidence for and discussion regarding that the fluorescence data during organelle contact event sites provides additional information from which any conclusions about the contact sites could be drawn would enhance this application.

Response. To address this, we performed additional experiments (“Smart acquisitions of mitochondrial divisions and constrictions” section). As the reviewer mentions, fluorescence imaging provides invaluable data for organelle identity (mitochondria, lipid droplets), but this might not require EDA. However, EDA is important for collecting information on organelle function (e.g. membrane potential), structures invisible in phase (e.g. nucleoids, DRP1), and rapid dynamics and fluctuations during the acquisition. Short-lived membrane potential variation (such as observed during fission but not upon vesicle contact) are fundamental for mitochondrial quality control. The rapid trafficking and reorganization of mitochondria and their nucleoids makes dynamic information critical for their accurate investigation. We are grateful for this feedback – these new analyses and textual changes better demonstrate the improvements of our imaging paradigm over traditional approaches for cellular health and dynamic resolution.

Q6. Page 4: It is mentioned that hybrid-EDA enabled imaging for “nine times longer”, since there was “one organelle contact site detected every 3.1 minutes on average”. I miss the calculation to support this claim, and cannot follow where the value “nine times longer” comes from. A clarification of this claim in the text would be helpful. The same goes for the $1/3.2$ minutes and 10x longer imaging that is mentioned for mitochondrial division events.

Response. We now include more details to clarify this estimation.

Added text [Results]. During labeling we identified one contact event on average every 3.1 minutes, with a duration of 20 seconds; thus, when imaging at 1 Hz hybrid-EDA captured 20 fluorescence frames per event instead of about 186 which would have been captured during continuous fluorescence. This represents a nine-fold reduction in phototoxicity per event, or a nine times longer accessible experiment duration.

With an estimated division rate of 1 per 3.2 minutes for a typical cell, when imaging at 1 Hz we could image ten times longer in hybrid-EDA by imaging 20 fluorescence frames per event, compared with about 192 if we were imaging continuously in fluorescence.

Q7. Page 6: When it comes to the experiments on mitochondrial division event sites, the same could be argued about the experiments where MitoTracker is the fluorescent target (Figure S6). In this case, the fluorescence data does not seem to add much information, as the resolution between phase contrast and fluorescence seems similar, and the tubular organelles are already distinctly identified as mitochondria, according to the discussion of the author about the organelle contact site event detection. A further discussion on what the fluorescence data adds in this case would be helpful. Experiments where DRP1 or TMRE are used as fluorescent targets instead presents clear benefits regarding the information one can gather from the experiments as opposed to pure phase contrast or fluorescence imaging.

Response. A main benefit in this data is that analysis tools for segmentation, network detection and morphology of mitochondria are typically developed with fluorescence data in mind. Those tools do not generally work well with phase contrast due to the additional signals from other cellular features. With this data, these established tools could be combined with the low phototoxicity of hybrid-EDA experiments. We have adjusted the caption on the supplementary Figure to explain this.

Modified caption Suppl. Fig. 8. Additional division events captured using hybrid EDA. A variety of mitochondrial morphologies triggered the adaptive fluorescence acquisitions in different cellular environments. Phase contrast imaging (top), MitoTracker fluorescence (bottom), and score obtained from the model on-the-fly as an image (crop marked as a red square at time 0). The event score is plotted below each series with a red line marking the threshold used. Capturing the fluorescence channel during the divisions allows for the use of established analysis tools for segmentation and quantification compared to the very scarce tool set available for phase-contrast data. This allows for analysis of mitochondrial length, diameter and network structure during division events (Supplementary Videos 9-12).

Q8. Page 6, first paragraph of Smart acquisitions of mitochondrial divisions and constrictions: Fig 3e is referenced, but it should be Fig 3f.

Response. Corrected.

Q9. Page 6: There is a claim that the mitochondria membrane potential is very sensitive to the stress state of the cell, here a reference would be good to support it.

Response. Added.

Added text [Results]. However, the membrane potential itself is very sensitive to the cell's stress state¹⁴ — meaning that even the moderate light exposure required for fluorescence imaging can induce changes.

Q10. Page 6: It would be beneficial to show more examples, for example in a supplementary figure, and statistics of the flickering of membrane potential at constriction sites – right now only one example is shown but in the text it is mentioned that multiple such events were captured. How often did you find such sites, and how many did you find in total? Furthermore, the example shown seems to not be connected to a division event, while the network was optimized to detect potential division sites. Is there any connection between reversible constrictions and membrane potential flickering, or does it also happen following division? How many events that you detected led to division and how many led to reversed constriction?

Response. We agree that this is an interesting correlation. We have analyzed more data like this and report the results in a new Supplementary Figure and the relative numbers in the main text.

Added text. Fluorescent dyes and proteins have developed to be increasingly efficient and photo-

stable; improvements in functional reporters are less forthcoming, perhaps because their specificity in sensing constrains optimizing other attributes beneficial for imaging. TMRE (Tetramethylrhodamine ethyl ester) is widely used to measure mitochondrial membrane potential. However, the membrane potential itself is very sensitive to the cell's stress state—meaning that even the moderate light exposure required for fluorescence imaging can induce changes. This sensitivity makes it challenging to reliably capture rapid fluctuations in membrane potential, especially those which accompany mitochondrial dynamics. We applied our smart imaging approach to measure membrane potential in the vicinity of constriction sites and at divisions and captured events of temporal membrane potential loss — or ‘flickering’ — associated with these sites. While whole-mitochondrion flickering is readily observed with traditional microscopy¹⁰, this rapidly reversible subdomain loss of potential would be undetectable at the lower imaging rates typically used in these experiments to avoid reporter photobleaching (Fig. 3h, Suppl. Fig. 6). Among a total of twenty division events, we captured 4 with significant fluctuations in membrane potential, only one of which was observed to reverse constriction without going to full separation of the organelles.

Q11. There is currently no reference to Fig S6/Supplementary Note 6 in the text – insert this at an appropriate point.

Response. Added.

Q12. In the Fig S6b legend it is stated that the F1 scores are improved with additional time points for all architectures and loss functions. However, from the graph it seems relatively clear that using the U-Net there is no significant improvement, neither with bce nor focal loss function. Was the comment mentioned for the stateful architecture only?

Response. Yes, for the stateful architecture – we adjusted the corresponding text accordingly (now Fig. S7).

Q13. Laser powers for fluorescence imaging, wherever mentioned, are mentioned in percentage. This arbitrary unit does not help the reader to understand what is going on, and is entirely subjected to e.g. the microscope and the microscope alignment at the day of experiments. This value should be replaced with a measure of power (W) or intensity (W/m²) at the sample plane, back focal plane or similar conjugate plane to ensure accurate reporting and allow any kind of comparison, especially since claims about sample health are made as a key point of the method.

Response. Agreed, in the revised section of the supplementary materials, we have replaced the percentage values with calibrated intensity values (mW/mm²) at the sample plane.

Q14. The data handling code provided (deep-events) has a limited README file with shallow installation instructions, further instructions would be helpful. Furthermore, it requires a running MongoDB server, and as this is not something I have available I cannot try out and review the code.

Response. We thank the reviewer for pointing this out. In the revised version, we have significantly expanded the README file to include detailed installation and usage instructions. Regarding the MongoDB requirement, we now provide guidance in the README on how to download and install MongoDB, with a direct reference to the official documentation.

Q15. As far as I understand the microscope control software structure, it works through the pymmcore-plus environment with a few additional programs running in synchronicity. For the modules that are not part of pymmcore-plus according to Fig S3, i.e. smartRunner, Analyser, Interpreter and Actuator, I cannot find references in the manuscript or supplementary information to any availability, and as such I am unable to review it. Accessibility to such code would be helpful and I expect that this code will be shared upon publication.

Response. We have added information and links to the specific components in the Supplementary Note 4. Many of the components are derived from the versions we used in the original event-driven acquisitions paper⁸. The exception is the new actuator that can switch between two independent acquisitions.

References

1. Chen, Q. *et al.* LW-DETR: A Transformer Replacement to YOLO for Real-Time Detection 2024. arXiv: 2406.03459 [cs.CV].
2. Dominguez Mantes, A. *et al.* Spotiflow: accurate and efficient spot detection for fluorescence microscopy with deep stereographic flow regression. *Nature Methods* **22**. cited By 1, 1495–1504 (2025).
3. Hattori, T., Watanabe, K., Uechi, Y., Yoshioka, H. & Ohta, Y. Repetitive transient depolarizations of the inner mitochondrial membrane induced by proton pumping. *Biophysical Journal* **88**. Cited by: 45; All Open Access; Green Accepted Open Access; Green Open Access; Hybrid Gold Open Access, 2340–2349 (2005).
4. Lee, H. & Yoon, Y. Transient contraction of mitochondria induces depolarization through the inner membrane dynamin opa1 protein. *Journal of Biological Chemistry* **289**. cited By 42, 11862–11872 (2014).
5. Li, B. *et al.* Equalized Focal Loss for Dense Long-Tailed Object Detection in 2022 IEEE/CVF Conference on Computer Vision and Pattern Recognition (CVPR) (IEEE, New Orleans, LA, USA, June 2022), 6980–6989.
6. Lin, T.-Y., Goyal, P., Girshick, R., He, K. & Dollár, P. Focal Loss for Dense Object Detection in 2017 IEEE International Conference on Computer Vision (ICCV) (Oct. 2017), 2999–3007.
7. Loew, L. M., Tuft, R. A., Carrington, W. A. & Fay, F. S. Imaging in five dimensions: time-dependent membrane potentials in individual mitochondria. *Biophysical Journal* **65**. Cited by: 206; All Open Access; Green Accepted Open Access; Green Open Access; Hybrid Gold Open Access, 2396–2407 (1993).
8. Mahecic, D. *et al.* Event-Driven Acquisition for Content-Enriched Microscopy. *Nature Methods* **19**, 1262–1267 (Oct. 2022).
9. O'Reilly, C. M., Fogarty, K. E., Drummond, R. M., Tuft, R. A. & Walsh, J. V. Quantitative Analysis of Spontaneous Mitochondrial Depolarizations. *Biophysical Journal* **85**. Cited by: 119; All Open Access; Green Accepted Open Access; Green Open Access; Hybrid Gold Open Access, 3350–3357 (2003).
10. O'Reilly, C. M., Fogarty, K. E., Drummond, R. M., Tuft, R. A. & Walsh, J. V. Quantitative Analysis of Spontaneous Mitochondrial Depolarizations. *Biophysical Journal* **85**, 3350–3357 (Nov. 2003).
11. Redmon, J., Divvala, S., Girshick, R. & Farhadi, A. You only look once: Unified, real-time object detection in. **2016-December**. cited By 42657 (2016), 779–788.
12. Santo-Domingo, J., Giacomello, M., Poburko, D., Scorrano, L. & Demaurex, N. OPA1 promotes pH flashes that spread between contiguous mitochondria without matrix protein exchange. *EMBO Journal* **32**. cited By 89, 1927–1940 (2013).
13. Stringer, C. & Pachitariu, M. Cellpose3: one-click image restoration for improved cellular segmentation. *Nature Methods* **22**. cited By 46, 592–599 (2025).
14. Zamzami, N. *et al.* Sequential reduction of mitochondrial transmembrane potential and generation of reactive oxygen species in early programmed cell death. *Journal of Experimental Medicine* **182**. cited By 1471, 367–377 (1995).

Response to Reviewers

Reviewer 3

We thank the reviewer for their careful evaluation of our rebuttal and their constructive comments. Below we provide a point-by-point response.

Q1 - Supplementary Movies I am unfortunately unable to see any modifications to the supplementary movies – the 13 files provided (labelled Supplementary Movie 1–13 and pertaining to figures 2e, 2f, 3f, 3h, 4a, 4b, 4c, 4d, 6a, 6b, 6c, 6d, 6e in the reviewer files) look to be identical as in the first submission. I can see references added to a few movies in the main text: “Movie” at the mention of Fig. 2e, “Movie” at the mention of Fig. 2f, “Supplementary Videos 9–12 in Suppl. Fig. 8 legend. The rest of the supplementary movies (3–8) are still unreferenced in the text, and it would be clearer if the mentions are uniform and precise (such as Suppl. Movie X), rather than just referring to “Movie”. The labels in the reviewer files where some seem to pertain to figures 4 and 6 is very unclear, as these figures do not exist. I assume that Suppl. Fig. 5 and Suppl. Fig. 8 are meant instead, but neither that adds up as Suppl. Fig. 8 does not have a panel “e”. I still believe the movies would be improved by adding channel labels, and potentially adding legends to the movies in the supplementary material. Seeing the rebuttal, I believe that maybe the authors have updated the movie files themselves, but that the updated files might not have been uploaded properly to the submission system?

Response. We apologize for the inconvenience caused and thank the reviewer for the careful assessment. In this revised submission, all supplementary movie files have been updated and uploaded to ensure full consistency with the manuscript. Specifically, we have:

- Ensured that all 16 supplementary movies are the updated versions, including the channel labels as suggested.
- Added clear and consistent references to all supplementary movies in the main and supplementary text (uniformly cited as Suppl. Movie X).
- Added legends for all the movies, in the new related section of the SI (p. 12, “List of Supplementary Movies”).

Q5 - Organelle contact sites fluorescence information. I appreciate the effort to perform additional experiments to show an application where the fluorescence data can be helpful. In the new Suppl. Fig. 6, it would be good to mention in the legend how many events were considered in the quantification in panel b. I believe there has additionally been an error in the ordering of the frames, where 8 s and 10 s come before 6 s.

Response. In this revised version, we have corrected the frame order in Supplementary Fig. 6a and added the requested quantification details in the legend. Specifically, the legend now includes the following text: “A total of 459 mitochondria were quantified across 16 movies (~13 frames each), of which 30 were LD⁺ and 429 were LD⁻.”

Q8 – Figure reference This figure reference is not corrected, it still refers to Fig. 3e rather than 3f.

Response. The figure reference has been corrected to Fig. 3f in the revised manuscript.

Q10 – Membrane potential flickering at constriction sites. In my original question, I asked about more examples shown for the membrane potential flickering at constriction and division sites, which is the topic of the paragraph. The data presented in the new Supplementary Figure referenced (Suppl. Fig. 6) refers to membrane potential at lipid droplet contact sites, which instead refers to an earlier paragraph in the text. I am not sure why this Supplementary Figure is referenced here, other than the fact that that experiment uses the same fluorescence labeling? I do still believe that showing further examples of the membrane flickering in correlation to the constriction and division sites mentioned in the statistics would make the point stronger for this application (just as with the additional data displayed in Suppl. Fig. 5 and Suppl. Fig. 8 for the other applications presented), and I do otherwise wonder if there is some specific reason as to not show additional data?

Response. In this revised version, the reference to Supplementary Fig. 6 (mitochondria–lipid droplet contact sites) has been moved to the appropriate section of the text. Additionally, we have included and referenced a new Supplementary Fig. 9 showing examples of mitochondrial membrane potential during mitochondrial constriction and division events, as requested.

Q11 – Reference to Suppl. Fig. S8. Despite the reply in the rebuttal, I cannot see any reference to Suppl. Fig. S8 (previous S6) in the text or supplementary material. Has this been missed to be added?

Response. The reference to Supplementary Fig. 8 has now been added in the relevant section of the manuscript and verified for consistency with the supplementary material.

Suppl. Fig. 4 is currently of very low quality in the supplementary file.

Response. A high-resolution version of Supplementary Fig. 4 has now been provided in the revised supplementary material.

The supplementary table is labelled as “Suppl. Table 0” in its legend, but 1 in the text.

Response. The supplementary table is now correctly labeled as Supplementary Table 1 in both the legend and the supplementary text.